# The COVID-19 Experience: Creativity as an Identity Attractor for Young People Facing the Developmental Challenges

**DOI:** 10.3390/ijerph19158913

**Published:** 2022-07-22

**Authors:** Luigia Simona Sica

**Affiliations:** Department of Humanities, University of Naples Federico II, 80133 Naples, Italy; lusisica@unina.it

**Keywords:** creativity, individual resources, identity

## Abstract

The study focuses on identifying the impacts of the COVID experience on young people and exploring whether, during the pandemic period, adolescents and young adults resorted to flexible and creative coping strategies, which may have served as resources. The participants consisted of 70 Italian freshmen (18 males and 52 females) aged 18 to 21, attending their first year of university. Adopting a narrative approach, we identified seven creativity functions and two interpretative factors, supporting the idea that creativity may have constituted a psychological resource for young people during the COVID-19 pandemic. Furthermore, the findings suggest that creativity can be configured as an identity attractor. Implications and future research directions are discussed.

## 1. Introduction

The COVID-19 pandemic can be categorized as a “historical change” [1,2]; that is, a change caused by historical events that impacted all the members of a particular group. These events are not always predictable or foreseeable, but they are shared with other members of the group. The peculiarity of these events is that, precisely due to their unpredictability and their novel and exceptional nature, people (individuals and social groups) do not have adequate psychological tools to cope with them. In other words, having not previously experienced such an event, people cannot activate specific coping strategies. People experiment; that is, they respond to such events in an immediate, spontaneous, and creative way to activate personal psychological resources that may or may not prove to be adequate for coping with the new event, since they have not yet been tested in this area. Sources of stress that disrupt everyday life can act as catalysts for individual change [3]. Not only that, but a certain amount of stress can lead to the acquisition of new skills [4]. In this sense, the psychological development of an individual is considered a dynamic, open system [5,6] in which individual and environmental characteristics interact and influence each other, causing the individual continuous processes of self-organization to rediscover the balance broken by the eruption of the unexpected event.

From the perspective of positive developmental psychology [7], it is important to understand if and how individual psychological resources may have helped individuals cope with the historical change linked to the pandemic. These resources can be defined as “transversal” to the type of event to be faced and can therefore be considered resources that can be used in various life experiences, favoring self-organization processes.

In this study, we focused on identifying the impact of the COVID experience on young people, exploring whether, during the pandemic period, they resorted to flexible, immediate, and creative coping strategies that may have served as self-organization agents.

### 1.1. Identity Formation from a Dynamic Systems Perspective: The Challenge of the Pandemic

Among the various existing research perspectives, the process of identity formation has been seen from a dynamic systems viewpoint [5,8,9,10,11], and it can be interpreted as a higher-order configuration resulting from stable self-organization processes [8,12]. Specifically, Kunnen [5] (p. 171) argues that “identity development is conceptualized as an iterative process, in which each context–person transaction is described as one iteration. Such a transaction can result in a fit between the existing commitments of the person and the context, or in a mismatch or conflict. A fit confirms the existing identity.” Kunnen [5] (p. 180) also explores the role of conflict in identity development, suggesting that “conflict is a trigger to identity change. Conflict is followed by an increase in exploration, and after a longer period of time, the strength of the existing commitment decreases, and change in commitment is observed.”

Therefore, in this study, we intended to explore the role of an unexpected experience (the COVID-19 pandemic) as a conflict trigger for identity and well-being.

A wide body of literature has emphasized that, to achieve identity, the contemporary world requires more flexible and more individualized personal skills from young people than were required in the past [13,14]. Thus, to define themselves, adolescents and late adolescents have to identify a personal life trajectory [15,16], cope with complex contextual factors [17], and be open to changing their life plans and life conditions [18]. In this scenario, helping young people overcome identity confusion is an important aim for developmental psychologists that requires the identification of resources to support optimal identity development in late adolescents.

According to Sica and Aleni Sestito [19], building identity in today’s young people is an open process that requires continuous exchanges with an evaluation of the real world. These, in turn, involve the use by the young person not only of the processes of exploration and commitment that we already know but also a considerable capacity for renewal, malleability, and flexibility [19,20]. These abilities allow young people to develop and modify themselves in a continuous process while maintaining a sense of self that does not derive so much from maintaining consistency with their previous commitments but rather from the use of creative skills and a tendency to continue along their life path with confidence and a desire for growth by actively responding to the needs of the context. All these skills together configure a sort of meta-skill [2]; that is, transversal and superordinate skills that are not specifically related to the tasks that young people have to face from time to time on their development path but yet allow them to tackle any task at any time.

### 1.2. Creativity as an Individual Resource 

Creativity is a widely debated scientific construct that has most recently been associated with identity, personality traits, and productive activities. However, creativity is a complex construct that has been investigated over the last few decades from many different points of view (e.g., process, product, and person) [21]. Despite its complexity, researchers agree on certain assumptions: for example, that creativity can be defined as the ability to generate ideas, insights, and solutions that are original, flexible [22,23], and effective [24], and that creative outputs result from cognitive flexibility (flexible and divergent ways of thinking) and cognitive persistence (persistent and systematic ways of thinking) [for a review, see [25]]. According to this perspective, it is not necessary to be a genius or an artist to be creative. Beghetto and Kaufman [26] and Kaufman and Beghetto [27] indeed proposed a taxonomy to analyze creativity that underlines the breadth of contexts and functions of creativity. We refer to *everyday creativity* (the little c of the Four C model; [27]), as it refers to the creativity that includes those actions that most of us engage in on a regular basis.

Furthermore, Metzl and Morrell [28] incorporate the role of personal creativity in processes of resilience when people are exposed to adversity, and Merrotsy [29] questions the tolerance of ambiguity as a trait of a creative personality.

In addition to these cognitive characteristics, creativity also implies emotional factors primarily related to creative personality dimensions [30]. All the cognitive abilities related to creativity develop during adolescence. More specifically, adolescence is the period when fluency and flexibility develop with distinct trajectories for divergent thinking and insight [30], while middle adolescence is the period of explorative thinking [31]. Starting with this evidence, one research approach has focused on the possible contribution of creativity to self/identity definition. Thus, creativity has been considered either a component or domain of identity [32] or a resource in terms of positive self-definition and support for identity achievement [33,34]. Barbot and Heuser [34] identified the role of creative processes in identity formation during adolescence. Starting with the humanistic view of identity as an outcome variable, they consider creativity an independent variable that can support identity through three facets of creativity: creative thinking, creative commitment, and creative expression [34].

Sica, Kapoor, and Ragozini [20] recently investigated whether and when creativity can be a resource for individual well-being and optimal identity development during late adolescence and young adulthood by including the valences of positive creativity (the use of the creative process to achieve something beneficial to oneself or to society) and negative creativity (the use of creativity to meet morally questionable goals).

With specific reference to the COVID-19 pandemic, Kapoor and Kaufman [35] (p. 5) proposed the hypothesis that “such creativity is an avenue to make meaning of current happenings.” Indeed, some evidence has been collected that explored creativity during the COVID-19 pandemic, which suggested that: the lockdown period facilitated everyday creativity [36,37]. Adults and older adults reported lower levels of loneliness on days when they were more creative than usual [38], and creative abilities helped people deal with lockdown and improved their well-being [39,40,41]. Research on the impact of COVID-19 on the processes of identity formation and creativity in young people is currently lacking.

### 1.3. The Current Study

The aim of this study was to explore the role of unexpected experiences (such as the COVID-19 pandemic) as possible conflict triggers for identity and well-being in young people. Adopting a narrative identity approach, we focused on identifying the impact of the COVID experience on young adolescents and young adults, exploring whether, during the pandemic period, they resorted to flexible, immediate, and creative coping strategies that may have acted as self-organization agents.

Creative individuals are remarkable in their ability to adapt to almost any situation and to make use of whatever is at hand to reach their goals [42,43]. For this reason, creativity can be considered a potent predictor of social problem-solving [44] and an “inherent latent power” present in each person [45].

We chose a narrative approach for several reasons. First, it places specific emphasis on individuals’ subjective assessments of their personal experiences and on the stories they tell about themselves [46]. Second, studying narrative accounts provides insights into personal interpretations of experiences that are not accessible through questionnaire measures alone [47]. Indeed, a primary assumption of narrative identity research is that the impact of experiences on peoples’ identity development depends not just on what happens to people but also on the narrative meaning that they assign to events [48]. Meaning-making has been proposed as one of the major processes through which identity is constructed [49].

Therefore, the current study aimed to answer the following research questions:

Q1. *Is creativity perceived as an individual resource?* Previous research [20] has shown that creativity does not have a unique value as a resource for well-being. We, therefore, expected that not all the subjects would report creativity as a psychological resource useful for coping with difficult times. However, we were interested in understanding if and when young people use it as a coping strategy in their difficult times and in exploring the role and function that creativity could have.

Q2. *Was creativity used as an individual resource to cope with the pandemic experience*? In accordance with research results [39,40,41] on the use of creative activities during the lockdown, we expected that, even for young people, creativity could serve as a resource for coping with stressful situations.

Q3. *What is the role of creativity in identity and well-being?* On the basis of the literature summarized above that links creativity to identity formation processes and underlines how creative activities support well-being [50], we hypothesized that creativity could represent, for young people engaged in their identity formation task, a self-organization agent, which helps them to explore in-depth, make commitments, and also give meaning, continuity, and direction to self-experience, even in difficult moments such as the pandemic.

## 2. Materials and Methods

### 2.1. Participants and Procedure

The participants were selected by convenience sampling from students in their first year of university studying social sciences (psychology and sociology) in a large Italian city (Naples). The sample consisted of 70 Italian freshmen (18 males and 52 females) aged 18 to 21 years (M = 19.65; SD = 2.28). They came from middle to high socioeconomic backgrounds, with at least one member of their family having higher education qualifications and middle-level employment. Everyone voluntarily participated in the study, with no financial incentive, and they were informed of their right to terminate their participation in this investigation at any time.

Participant recruitment involved two main steps. First, a graduate trainee in psychology invited students during class time, calling for freshmen to take part in a study on general issues about the transition to adulthood in Italy. Subsequently, a psychologist familiar with the topics of the study and with whom the participants had no prior relationships collected the narratives during class time.

During the data collection, the psychologist provided the participants with information about the study, its goals and objectives, and progress. The subjects were informed that their data would be processed and stored following the requirements of the Personal Data Protection Code. The participants were provided with an open-ended questionnaire, which they completed during the sessions, and the duration of the process was not limited. The subjects were able to express their agreement or refusal to participate in the study by completing the questionnaire and marking one of the possible answers at the beginning of the questionnaire: “I agree to participate” or “I disagree to participate.” Participation was in line with the ethical standards of the 1964 Helsinki Declaration and its later amendments or comparable ethical standards, and the research obtained authorization from the local ethics committee.

The data were collected in September 2021 to investigate the impact of the pandemic experience on the participants’ autobiographical memories during a period of gradual post-pandemic reentry. All the respondents were assigned pseudonyms and are referred to by this pseudonym when they are quoted to guarantee confidentiality.

### 2.2. Materials

We adopted an event-focused narrative approach [51] that uses a written prompt to elicit narratives regarding real-life experiences constructed as part of the larger life story, with emphasis on a particular theme. The narrative measure consisted of two alternative prompts: “Think about a time when creativity helped you get through a difficult period or experience. Write about that time” and “I have no episode to tell. Creativity has never helped me.” The participants were asked to write a personal narrative in response to one of the two prompts. The time required to complete the measure ranged from 35 to 50 min. Approximately 90% of the individuals who were contacted completed the narrative; the other 10% of the students declined to participate for reasons unknown to us.

### 2.3. Narrative Coding Procedure

We analyzed the narratives using content analysis and the guided multiple reading approach for interviews [52,53] adapted to identity narratives [54,55]. In every study step, the unit of analysis adopted was the whole narrative of each participant; thus, every narrative coincides with a participant, and the narrative corpus consisted of 70 texts. In the first step, each narrative was classified according to whether it was a narrative of creativity as a resource (yes/no). Each narrative was read several times, bearing in mind the different concepts regarding both creativity and identity dimensions. Passages in the text with content bearing on each concept were highlighted for future observation.

In the second step, each narrative was analyzed separately as a whole, codifying the text according to the coding system developed by the author in light of the literature described before (see Table 1). In this step, and in all the other steps of the coding procedure, two researchers coded all the data, with inter-rater reliability calculated based on the entire set; all reported reliabilities are deemed within the acceptable range [56].

Specifically, each narrative was coded using a bottom-up approach in terms of the functions of creativity and meaning-making (see Table 1). In order to define meaning-making, the system to examine meaning-making from a developmental perspective developed by McLean and Thorne [57] was used (see also [58]). According to this system, meaning-making processes reflect increasing complexity in autobiographical reasoning, starting from a lesson learned, defined as meanings that are often behavioral and do not extend the meaning beyond the original recalled event, and progressing to gaining insight, defined as meaning that extends beyond the specific event to explicit transformations in one’s understanding of oneself, one’s relationships, or the world. Thus, the meaning-making processes were coded using McLean and Pratt’s [58] system with increasing scores of complexity in autobiographical reasoning.

The third step focused on identifying the recurring associations in all the categories identified in the previous steps and on identifying interpretative factors. To do this, we first re-aggregated the narratives, grouping them by the coexistence of the same level of meaning-making and the same categories of creativity (both identified in the second step of analysis). As a final step, we reread the narratives within each aggregated group, trying to trace only the psychological processes described and the narrative functions beyond their context of use (e.g., relational, scholastic). We defined this level as the identification of latent interpretative factors. At this point, we were able to re-aggregate the narratives according to the latent interpretative factors that emerged. It should be noted that, as Bryman [59] states, regarding the issue of the external validity of qualitative studies, ‘‘the issue should be couched in terms of the generalizability of cases to theoretical propositions rather than to populations.’’ Furthermore, this research attempted to demonstrate the plausibility of different typologies and not to assess their relative frequencies in the general young Italian population.

Finally, three supplemental checks to enhance the trustworthiness of the findings and the methodological integrity of our qualitative study were used: asking participants whether they felt that the narrative was complete and exhaustive of the theme; data displayed in Excel format to aggregate and vividly portray the narrative findings; and narrative examples to relate the findings to the participants’ real experience.

## 3. Results

First, in terms of creativity as a resource, 87.5% of the students described the use of creativity as an individual psychological resource, while 12.5% stated that they had never resorted to creative strategies. 

Second, the respondents reported seven categories of creativity functions as follows (see Table 2): cope with difficult times (42.6%), conduct tangible problem solving (18.5%), withstand COVID-19 lockdowns (14.81%), engage in self-expression (12.9%), help others (9.26%), benefit from the creativity of others (1.85%), and engage in narrative meaning-making (1.85%). The average level of meaning-making in the narratives was M = 1.45, SD = 0.63, range = 0–3, indicating some presence of meaning-making on average, but with wide variability across participants.

Third, making relationships between the meaning-making level and the different creativity as a resource modality, we aggregated the different narratives across the sample to create the interpretative creative modalities. According to the co-occurrences of the level of meaning-making and creativity functions, the coders defined a first grouping (a, b, c) for defining the modalities. Subsequently, we returned to qualitative analysis by rereading the narratives to explore in-depth the groups’ characteristics. Following this procedure, the coders continued with the progressive qualitative analysis of the groups up to the saturation of the co-occurrences and until there were no further differences. Finally, three groups or modalities were identified and labeled as follows: *creativity as attractor*, *creativity as support*, and *creativity as solution* (Figure 1).

*Creativity as attractor*: Both in the descriptions of creativity used as a resource to cope with the pandemic and in the descriptions of creativity used to overcome difficult moments, a common characteristic was the description of behavior that was repeated over time (e.g., drawing every day; writing whenever you are unable to make sense of experiences, emotions, and parts of yourself). This behavior catalyzes thoughts and actions and “absorbs” the subject’s attention with comforting routines that help the person through difficult moments. This characteristic is reminiscent of the definition of “attractor” applied in the psychological sciences: “a state—a specific place in the state space—to which the system tends to return” [60] (p. 22). It is for this reason that the functions “cope with difficult times” and “withstand COVID-19 lockdowns” are aggregable, as they both constitute an attractor for the psychological well-being of the person who uses them. That is, in this case, creativity becomes a stable configuration to which the young person returns whenever he or she has difficulty facing internal or external challenges.*Creativity as support*: The narratives reporting experiences of helping other people constantly referred to creativity as a resource that was useful in developing a new behavioral strategy to help other people. In these narratives, creativity is described as the function of support in helping others, or as a tool to find the right way to provide help and care.*Creativity as solution*: This creativity typology emerged in the narratives belonging to both “engage in self-expression” and “conduct tangible problem-solving.” It is related to classical problem solving, with a direct reference to practical, tangible, and concrete problems. In both cases, people described behaviors and actions directed toward finding a solution to either a concrete, external problem (e.g., repairing a broken piece of furniture while on vacation) or to individual issues of self-expression (trying a new hair color or creating a movie by mixing images and words, etc.). The value of creativity is to identify a solution, restore a previous situation or balance, or find a new one, but the focus is on the solution.

A subsequent reading of the narratives of each profile contributed to defining latent dimensions for all narratives and semantic differences between the profiles regarding the description and interpretation of the identified latent dimensions. Specifically, we identified two processes underlying the three interpretative creative modalities, regardless of their context of use, which we labeled as stabilizing and identity attractors (Figure 2). 

In the support and solution modalities, the respondents described the way they used a process to create a solution for a problematic situation for which they felt responsible. The problem was one that needed to be solved or “repaired” to allow the person to proceed; we have defined this as a “stabilizing” factor (e.g., one that allows the person to overcome the crisis and restore the previous equilibrium).

In the modalities with medium-high meaning making (attractor), we identified in the respondent’s narratives specific references to the processes of identity formation, both in terms of self-understanding (identity understanding) and in terms of in-depth exploration of the person’s own possibilities, as well as references to a commitment to new activities to overcome the challenge (identity challenge) of the changed living conditions during the lockdown. In addition to identity understanding and identity challenge, a specific function was identified in the second phase of the coding that was linked to the most expressive components of identity (“engage in self-expression”). We labeled this comprehensive factor “identity attractor.” 

## 4. Discussion

Prior to this investigation, research has suggested that creativity can be an intangible resource for well-being and the optimal development of identity [20,50]. Following this suggestion, in this study, we explored the hypothesis that creativity may have constituted a psychological resource for young people during the COVID-19 pandemic too. Adopting an interpretative perspective based on the model of dynamic systems, it was possible to figure out that the pandemic experience represented a real historical event that individuals were unprepared to cope with. In general, young people did not have previous experience dealing with such an event. Thus, focusing our attention on university students (who are already immersed in the dynamic processes of building their own identity), we asked whether the pandemic experience constituted a trigger in the process of identity formation and whether creativity was used as a coping resource. We formulated three research questions to understand (1) whether creativity was perceived as an individual resource in general, (2) whether creativity was used as a resource to deal with the specific pandemic experience, and (3) what the role of creativity was in the identity and well-being of the young participants in the research.

The results of our narrative study confirmed our hypothesis and provided an unexpected result that allowed us to propose an interpretative model linked to the functions of creativity as a resource.

First, in terms of Q1, most of the respondents reported using creativity (in the broadest sense of the term) as a resource in their lives. These data allow us to assume that creativity can actually be used as a psychological resource. Specifically, considering creativity as an individual psychological resource for well-being, we identified seven functions (i.e., cope with difficult times, conduct tangible problem-solving, withstand COVID-19 lockdowns, engage in self-expression, help others, benefit from the creativity of others, and conduct narrative meaning-making); three modalities (creativity as an attractor, creativity as a social support, and creativity as a tangible or expressive solution); and two factors (stabilizing and identity attractor).

Regarding Q2, about 15% of the respondents said that they had used their creativity specifically to cope with the period of the lockdown during the pandemic. These data indicate that creativity helped young people endure the lockdown, but it also shows that creativity can be used as a transversal resource for individual well-being. This finding is in line with research on adults implemented in other cultural contexts [36,37,38,39,40,41], which underlines the role of creative activities in coping with lockdown conditions. Our findings add to previous results transversally in the fields of use of creativity (not only coping with pandemics, but with many other difficult experiences).

This assumption is evidenced by the different types of use, or in other words, by the different functions of creativity that have been described, in addition to the use of creativity in coping with the pandemic. Specifically, this study identified seven different creativity functions. Among these, “conduct narrative meaning-making” and “benefit from the creativity of others” are relevant for adding new support to theory in the broad creativity field. Specifically, “conduct narrative meaning-making” supports the precise hypothesis of Kapoor and Kaufman [35] that creativity may have been used with a function of meaning-making during the lockdown. However, our results also show that not only during lockdown did creativity take on this function: our respondents described the use of narration (writing, novels, diaries, poems, and musical texts) precisely with the aim of understanding the meaning of difficult experiences in the wide course of their lives. Furthermore, in addition to supporting the idea that creativity can have a meaning-making function, our findings also support the vast literature on the narrative function of activating meaning-making, as well as cognitive and emotional restructuring [57,58,61,62,63].

With regard to the function of “benefit from the creativity of others,” it was minor in quantitative terms but interesting from a qualitative point of view. This function recalls the specific meaning of creativity as an artistic product and refers to the importance of the use of art as a balancer of internal emotional states. This perspective needs to be explored in future qualitative and quantitative studies. If, on the one hand, this result is in line with recent literature that suggests that the arts can promote health and psychological well-being and offer a therapeutic tool for many (e.g., adolescents, elderly, and vulnerable individuals [64,65,66]), on the other hand, it opens up new interpretative possibilities with respect to the non-momentary function of artistic fruition and to its habitual use as a coping strategy for stressful events and self-regulation of internal emotional states. This line of research seems particularly interesting.

Subsequently, by crossing the levels of analysis and investigating the relationship between meaning-making processes emerging from the narration of experiences and identity functions (Q3), three interpretative creative modalities emerged (creativity as attractor, creativity as social support, and creativity as a tangible or expressive solution). Proceeding in the work of identifying latent interpretative factors, we reread the narratives by identifying the processes underlying the three interpretative creative modalities, regardless of their context of use, and we identified two latent interpretative factors: *stabilizing* and *identity attractor*.

The identification of these two factors, in our opinion, is of particular interest and points the way toward future studies. The evidence that creativity is used in daily life as a resource to respond to different types of challenges confirms that creativity may be an important factor in individual well-being. Moreover, creativity, as it emerged in our study, is a strategy that young people spontaneously use to meet challenges. Our data also show that the use of specific creative methods depends on different underlying factors and thus functions differently in different subjects. For example, creativity may act as a stabilizer and almost an emotional regulation strategy with a prevalent behavioral component with the explicit goal of overcoming a challenge. Second, creativity seems to behave as an attractor for the processes of identity formation, since it manifests as a self-organizing process with an adaptive function. The emergence of these two factors (which are interpretative and, therefore, must subsequently be proven with a specific and in-depth line of research, which we are about to undertake) also provides an agenda for identifying identity attractors that could provide a new perspective on the study of identity.

### Limitations and Future Directions 

This study has several limitations. The first limiting element to be recognized for this study resides in its character of study focused on a group of individuals self-selected for voluntary participation and interest in the research topics. The study, therefore, has a preliminary nature and is configured as a pilot study. Alongside the sampling that can be expanded to increase the validity of the research, it should be emphasized that the data analysis methodologies used in this study represent only one of the existing approaches for the analysis of textual data. A comparison with other analysis strategies could deepen the evaluation of the subjects’ responses and support the research results in a divergent/convergent way. 

In particular, the study was exploratory and focused on a group of participants at a single point in time. Therefore, longitudinal and/or cross-sectional research is needed to support a more specific set of conclusions regarding the role of creativity in identity formation. Specifically, narrative longitudinal research would facilitate the study of the role of creativity as an identity attractor in relation to personal experiences during the transition to adulthood. Second, the present findings can be applied only to university students. The creativity and identity experiences of youth who are not attending college may be different, and this could be the focus of future research. Third, our qualitative analysis resulted in a list of seven creativity functions; this list is not necessarily exhaustive and is offered as a basis for further research and exploration. Furthermore, future research may employ different data interpretation perspectives (e.g., a classical perspective in terms of the person, process, and product components of creativity).

## 5. Conclusions

Based on the preliminary and exploratory data of the present study, we conclude that creativity was a psychological resource during the pandemic lockdown for the well-being of the adolescents and young adults who participated in our study and that creativity can be considered a transversal resource for individual well-being [18,19] that can constitute an *identity attractor*. This has an applicative value at several levels. First of all, at an applicative level, it is confirmed that using creative activities can help well-being even in young people; these data, therefore, are added to the already existing evidence for adults and the elderly supporting the use of creativity as a resource for psychological well-being. Starting from this evidence, it may be desirable that not only creative activities are used in support interventions for adults (especially the elderly) with difficulty, but that they are also supported and implemented with young people to cope with developmental stressful situations. The pandemic period, as emerged from our data, represented an example of a field of application. However, from the students’ narratives, it emerged that creativity helped many young people to cope with stressful situations of various types (relational, mourning, concrete). This leads us to recommend to developmental psychologists that the enhancement of creative strategies should be understood as a support for the formation of transversal or meta-individual psychological skills. Among these, for the life cycle period studied in this research, the task of defining identity is of particular importance. It represents, on the one hand, a challenge for students and, on the other hand, an essential focus for their well-being. It is on this last point that the theoretical implications of our study rest, which, as we have seen in detail previously, show that creativity can constitute a real identity attractor.

This latter interpretation will be tested and confirmed with appropriate quantitative/qualitative, longitudinal, and subject-centered research methodologies, but at present, based on the narrative data of our study, we conclude that the COVID-19 experience has revealed creativity as an identity attractor in the achievement of well-being.

## Figures and Tables

**Figure 1 ijerph-19-08913-f001:**
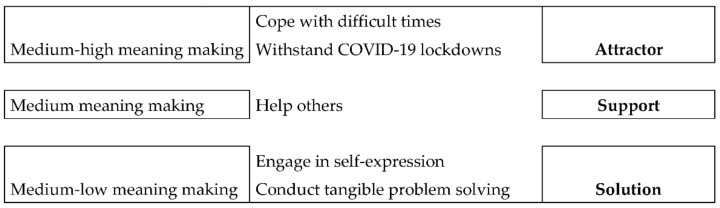
The interpretative creative modalities: attractor, support, and solution.

**Figure 2 ijerph-19-08913-f002:**
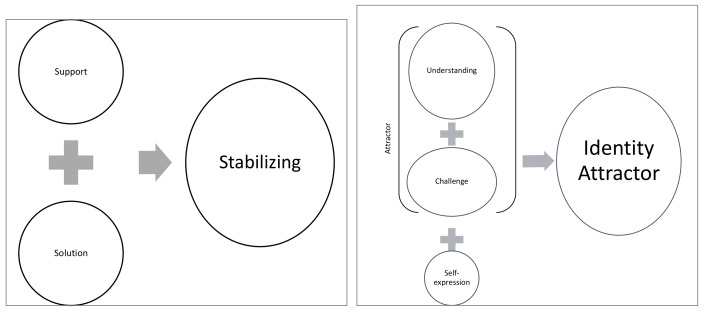
The interpretative factors: Creativity as a stabilizing factor and as an identity attractor.

**Table 1 ijerph-19-08913-t001:** Coding system.

Research Question	Category of Analysis	Description
Q1. Is creativity perceived as an individual resource?	Yes/No	Response to the narrative prompt
Q2. Was creativity used as an individual resource to cope with the pandemic experience?	Presence of experience related to COVID	Content analysis
Q3. What is the role of creativity in identity and well-being?	Meaning-making level Interpretative creative typologies	Scored 0–3, with 0 for narratives with no meaning reported and 3 for narratives with insights, which were defined as meanings that extend beyond the specific event

**Table 2 ijerph-19-08913-t002:** Creativity as a resource: Descriptive results.

Emerged Categories Creativity Functions	Description Narrative Examples
1. Cope with difficult times	*Creativity has helped me in many difficult times. When my father had an accident and was hospitalized for weeks, I spent the days drawing. During the most difficult period for me in high school, I started writing poetry. When I am sad, I take what I find and do cosplay or create a scenario; I analyze the topic in my mind … Producing an idea, creating something (tangible or not) is the best way I know to feel better when a difficult time is going on.*
2. Conduct tangible problem-solving	*When I was on vacation with my friends in the summer, we ended up breaking a sofa bed in the apartment we were staying in, and we didn’t know how to fix it. We searched the Internet but found nothing, so it occurred to me to look for glue to repair the damage, but as it was Monday, all the hardware stores were closed. I did not lose heart and had the intuition to look for it in a botanical shop, and there we found it. So we were able to buy it and avoid the problems*.
3. Withstand COVID-19 lockdowns	*During the first lockdown, I was sad and very thoughtful; closed up at home, I could not escape thoughts and questions whose only solution was to find answers. And I found them by combining movie scenes I saw with phrases from books I read along with music lyrics I listened to. This allowed me to draw my own thoughts and ideas, which acted as answers (albeit not totally conclusive) to the questions that tormented me, mainly due to the quarantine.*
4. Engage in self-expression	*Creativity is not constantly active in me, that is, there are moments in which I feel particularly inspired and, therefore, suddenly, I jot down a few words that can express my thoughts, state of mind. The poem made me feel “free” from what I had inside, both positive and negative.*
5. Help others	*In high school, I had a classmate with difficulties in independent learning. Outside school hours, the support teacher could not assist her, and sometimes we studied together. Having a slight delay in her learning, it was not easy for her to use common learning strategies, so creativity helped me help her.*
6. Benefit from the creativity of others	*It helped me get through almost all of my tough times. In moments when I didn’t feel listened to and couldn’t listen to myself, she allowed herself to see me. She always helped me to let go of what was accumulating inside, and she helped me understand that I continued to exist (that I was alive) even when I felt non-existent. She pushed me to get involved.* *The creativity of others helped me a lot, seeing the words, images, and drawings of others in the moments when I felt I could not see them or have them myself. The creativity of others pushes me so much to cultivate my own.*
7. Conduct narrative meaning-making	*It was a difficult time as I felt confused and overwhelmed by my surroundings and had the perception that I could not do anything well. Using creativity, I have created something that is nonsense, but it has allowed me to put my thoughts in order.*

## Data Availability

The data that support the findings of this study are available on request from the corresponding author. The data are not publicly available due to their containing information that could compromise the privacy of research participants. The data are available from the corresponding author on reasonable request (lusisica@unina.it).

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
