# Peer review of "The COVID-19 Experience: Creativity as an Identity Attractor for Young People Facing the Developmental Challenges"

_ijerph, 2022, doi:10.3390/ijerph19158913_

Round 1

Reviewer 1 Report

The article can be enhanced with more references to similar studies in different cultural and social contexts.

Author Response

R1: The article can be enhanced with more references to similar studies in different cultural and social contexts.

REPLY: The comparison with studies carried out in other cultural contexts is certainly at the basis of the analysis of literature that was necessary to support this study. I thank Refree for indicating this point as still lacking. I have therefore expanded the references to international literature included in the introductory and theoretical part of the paper (pages 2 and 3). Obviously I have added the relative 10 bibliographic references in the References section.

Reviewer 2 Report

This paper shows that creativity contributes to coping in a pandemic. How has previous research dealt with this relationship surrounding creativity? Please refer to more papers to clarify how this paper is positioned within them. What is the meaning of this paper in the real world? For example, what kind of measures can be applied in educational administration? The first half of the conclusion is the limitation. The limitation should be moved to the limitation clause. Then, the central discussion of this paper should be developed in conclusion.

Author Response

I thank the Reviewer for his inspiring comments which allowed me to improve the quality of the study. We have tried to answer all your requests, as follows:

R2: how has previous research dealt with this relationship surrounding creativity? Please refer to more papers to clarify how this paper is positioned within them.

REPLY: We have added in the introductory section, more extensive references to the literature that forms the background to our study, specifying in more detail the research on creativity linked to our approach (p. 2 lines 86-94). We also added a paragraph that explores the specific theme of the link between creativity and Covid, reporting about ten studies that currently investigated this topic (p.3 lines 113-121).

R2: What is the meaning of this paper in the real world? For example, what kind of measures can be applied in educational administration? The first half of the conclusion is the limitation. The limitation should be moved to the limitation clause. Then, the central discussion of this paper should be developed in conclusion.

REPLY: I am very grateful to the reviewer for soliciting an in-depth study of the application use of the results of this study. I have in fact expanded this section in the Discussion (text in green in the manuscript). Also with reference to the observation with respect to the structuring of the Conclusions, I have separated the paragraphs conclusion and limitations/future directions, thus giving greater clarity to the discussion of the results.

Reviewer 3 Report

The study aims to show that creativity in first-year college students was an asset during the COVID-19 pandemic.

The study is of a narrative nature. This type of approach requires compliance with some essential conditions. First, that the concept under consideration is firmly established in the literature. The second, that the sample examined is representative of the population.

1.       The young researcher cited seven of her own studies. This testifies to her interest in the subject but does not give the reader the opportunity to correctly frame this research within the scientific literature on the subject. Creativity in positive psychology, especially student creativity, has attracted thousands of publications. The author should try to synthesize the literature, taking into account the different trends.

2.       The author state that “The greater presence of women in our student sample reflects the proportion of women in Italian universities” (Line 133). Data from the Italian University Ministry indicate that in 2022 there are about 23,000 students and that males and females are in equal proportions. I therefore suggest that the researcher balances the sample by recruiting an equal number of male students.

3.       It would also be important for the researcher to explain which faculty she drew her sample from. If only first-year psychology students were interviewed, there is no evidence that they are the same as students from other faculties.

4.       The method was developed by the researcher herself. In this case a comparison with other methods would be useful.

5.       The researcher reports that participation was voluntary and unpaid. She should also tell you how the recruitment was carried out. Since all the interviews took place in just one month, it would be useful to know the length of the interviews.

6.       It is hardly necessary to observe that, if the students were enrolled in the first year of the degree course and the author was the teacher of the course, this information should be made available because it is potentially able to significantly modify the answers in a socially desirable sense.

Author Response

R3: The study is of a narrative nature. This type of approach requires compliance with some essential conditions. First, that the concept under consideration is firmly established in the literature. The second, that the sample examined is representative of the population.

REPLY: Thank you to the anonymous reviewer who greatly helped us to improve the paper, thanks to their very detailed comments. I have done my best to clarify the points evidenced by modifying the text, clarifying the structural, methodological and theoretical aspects as required. My notes to each concerns/suggestions are anticipated by REPLY. For each main answer, I have indicated the page/pages of the revised manuscript in which is possible to see how we have changed the paper accordingly to the concerns/suggestions we have received.  We marked the most important changes in the manuscript using green colour for the changed text.

  1. The young researcher cited seven of her own studies. This testifies to her interest in the subject but does not give the reader the opportunity to correctly frame this research within the scientific literature on the subject. Creativity in positive psychology, especially student creativity, has attracted thousands of publication The author should try to synthesize the literature, taking into account the different trends.

REPLY: I thank the reviewer for stimulating an in-depth look at the scientific literature. Surely the literature on creativity is vast and has been produced in various fields of psychology and beyond. However, I agree with the referee that the paper needed a clearer anchorage to the specific reference line. On the other hand, some extension of the references was also requested by Referee 1. Therefore, I have expanded the references to international literature included in the introductory and theoretical part of the paper (pages 2 and 3). Obviously, I have added the relative 10 bibliographic references in the References section.

  1. The author state that “The greater presence of women in our student sample reflects the proportion of women in Italian universities” (Line 133). Data from the Italian University Ministry indicate that in 2022 there are about 23,000 students and that males and females are in equal proportions. I therefore suggest that the researcher balances the sample by recruiting an equal number of male students.

The suggestion of sample balancing is certainly interesting and would improve the possibility of making gender assumptions in future studies similar to the one presented. Now, I am afraid that modifying the sample object of this study could also modify the meaning making processes. The narratives referred to in this study were in fact collected in September 2021 to investigate the impact of the pandemic experience on the partici-pants' autobiographical memories during a period of gradual post-pandemic reentry. Widening the sample now would lead to comparing different narratives in which memory probably acted in a reconstructive way on the evaluation of experiences. However, it would be interesting to expand the study, in a longitudinal way, by balancing the sample in the following steps. In any case, in the present study we have not taken into specific hypothesis linked to gender variable.

  1. It would also be important for the researcher to explain which faculty she drew her sample from. If only first-year psychology students were interviewed, there is no evidence that they are the same as students from other faculties.

REPLY: Thanks for the in-depth suggestion. The participants paragraph has been rewritten introducing more detailed information on participants and the recruitment procedure. This is also to dispel the doubts legitimately advanced by the referee regarding the possible conflict in the data collection conducted by the teacher.

  1. The method was developed by the researcher herself. In this case a comparison with other methods would be useful.

REPLY: I thank the reviewer for giving the opportunity to clarify this point. The data analysis procedure used has precedent in the literature not only of the author of this paper, but has previous bases in other references in the international literature. Surely, the initial presentation method of this procedure could generate misunderstandings. This is why the paragraph on the narrative data analysis procedure has been modified and integrated in its entirety, to provide the greatest number of useful indications for assessing the integrity and transparency of the method used ( p. 5).

  1. The researcher reports that participation was voluntary and unpaid. She should also tell you how the recruitment was carried out. Since all the interviews took place in just one month, it would be useful to know the length of the interviews. It is hardly necessary to observe that, if the students were enrolled in the first year of the degree course and the author was the teacher of the course, this information should be made available because it is potentially able to significantly modify the answers in a socially desirable sense.

REPLY: As previously indicated, we have rewritten the participants and procedure paragraph by entering the requested information (on p. 4).

Reviewer 4 Report

The authors made an interesting work. However, misses important points. 

Introduction lacks of theories (behavior) to support the information. The English is hard to follow. Also there's a lack of references along the introduction.

Creativity subsection: OK.

The study: based on the research question the authors may hilight the objective and the hypothesis.

Methods:

Ok. But the narrative approach miss information about validity. 

Results:

Ok

Discussion:

There are no references in discussion (?). Misses of explanation of the results and the authors did not compare or explain what they found with the l iterature. 

Author Response

R4: Introduction lacks of theories (behavior) to support the information.

REPLY: The comparison with studies carried out in other cultural contexts is certainly at the basis of the analysis of literature that was necessary to support this study. I thank Referee for indicating this point as still lacking. I have therefore expanded the references to international literature included in the introductory and theoretical part of the paper (pages 2 and 3). I have added the relative 10 bibliographic references in the References section.

R4: The English is hard to follow.

REPLY: The manuscript has been entirely re-edited and corrected by a native speaker

R4: Also there's a lack of references along the introduction.

REPLY: The number of bibliographic references, relating to international literature, added in the introduction and in the Discussion is equal to 22 studies

R4: The study: based on the research question the authors may hilight the objective and the hypothesis.

REPLY: In accordance with the Reviewer's requests, we investigated each research question, also supporting it with bibliographic references (pp-3-4). 

R4: Methods Ok. But the narrative approach miss information about validity. 

REPLY: I thank the reviewer for requesting an extension of this fundamental aspect for qualitative research. Also according to Reviewer 3, the paragraph related to the data analysis section has been completely rewritten, as well as that related to the participants and the sample recruitment procedure. Overall, an attempt was made to provide as much information as possible to guarantee the integrity of the work. 

R4:There are no references in discussion (?). Misses of explanation of the results and the authors did not compare or explain what they found with the l iterature. 

REPLY: In accordance with the reviewer's suggestion, the Discussion was remodeled and integrated with elements of comparison with the international literature and elaboration on the results that emerged. The most significant changes are shown in green in the manuscript (pp- 10-11)

Round 2

Reviewer 2 Report

The author made appropriate improvements.

Author Response

Thanks to the reviewer for allowing us to improve the manuscript with her/his valuable suggestions.

Reviewer 3 Report

Although the author has tried to improve the manuscript text, the study remains very poor and marred by numerous biases. The author should have discussed these limitations in the appropriate section of the manuscript; the section on limitations, on the other hand, remained unaddressed.

Author Response

Thanks to the reviewer for allowing us to improve the manuscript with her/his valuable suggestions. We hope that in subsequent studies we will be able to improve the research topic providing more consistent results.
As suggested by the reviewer, we have expanded the limitations section of the paper to report problematic aspects that were probably not sufficiently developed in the previous version (in red in the text).

Reviewer 4 Report

The authors have made a improvements in the manuscript.

Author Response

(The authors gave the same response as above.)
